# Simultaneous Isolation and Identification of Largemouth Bass Virus and Rhabdovirus from Moribund Largemouth Bass (*Micropterus salmoides*)

**DOI:** 10.3390/v14081643

**Published:** 2022-07-27

**Authors:** Yuqi Jin, Sven M. Bergmann, Qianyi Mai, Ying Yang, Weiqiang Liu, Dongli Sun, Yanfeng Chen, Yingying Yu, Yuhong Liu, Wenlong Cai, Hanxu Dong, Hua Li, Hui Yu, Yali Wu, Mingjian Lai, Weiwei Zeng

**Affiliations:** 1Guangdong Provincial Key Laboratory of Animal Molecular Design and Precise Breeding, School of Life Science and Engineering, Foshan University, Foshan 528231, China; vetyqjin@126.com (Y.J.); 2112159088@stu.fosu.edu.cn (Q.M.); yy41613@fosu.edu.cn (Y.Y.); 2112159087@stu.fosu.edu.cn (W.L.); 2112159097@stu.fosu.edu.cn (D.S.); chyfwf@126.com (Y.C.); yuyin123@fosu.edu.cn (Y.Y.); flora1988@scsio.ac.cn (Y.L.); 2112059069@stu.fosu.edu.cn (H.D.); okhuali@fosu.edu.cn (H.L.); yu71hui@aliyun.com (H.Y.); 2Institute of Infectology, Friedrich-Loffler-Institut (FLI), Federal Research Institute for Animal Health, Südufer 10, 17493 Greifswald-InselRiems, Germany; sven.bergmann@fli.de; 3Department of Infectious Diseases and Public Health, Jockey Club College of Veterinary Medicine and Life Sciences, City University of Hong Kong, Kowloon, Hong Kong 999077, China; wenlocai@cityu.edu.hk; 4Foshan Institute of Agricultural Sciences, Guangdong, Foshan 528145, China; yaliwoo2007@126.com (Y.W.); lmj-006@163.com (M.L.)

**Keywords:** *Micropterus salmoides*, largemouth bass, co-infection, largemouth bass virus, hybrid snakehead rhabdovirus, virus isolation, virus characterization

## Abstract

Largemouth bass is an important commercially farmed fish in China, but the rapid expansion of its breeding has resulted in increased incidence of diseases caused by bacteria, viruses and parasites. In this study, moribund largemouth bass containing ulcer foci on body surfaces indicated the most likely pathogens were iridovirus and rhabdovirus members and this was confirmed using a combination of immunohistochemistry, cell culture, electron microscopy and conserved gene sequence analysis. We identified that these fish had been co-infected with these viruses. We observed bullet-shaped virions (100–140 nm long and 50–100 nm in diameter) along with hexagonal virions with 140 nm diameters in cell culture inoculated with tissue homogenates. The viruses were plaque purified and a comparison of the highly conserved regions of the genome of these viruses indicated that they are most similar to largemouth bass virus (LMBV) and hybrid snakehead rhabdovirus (HSHRV), respectively. Regression infection experiments indicated fish mortalities for LMBV-FS2021 and HSHRV-MS2021 were 86.7 and 11.1%, respectively. While co-infection resulted in 93.3% mortality that was significantly (*p* < 0.05) higher than the single infections even though the viral loads differed by >100-fold. Overall, we simultaneously isolated and identified LMBV and a HSHRV-like virus from diseased largemouth bass, and our results can provide novel ideas for the prevention and treatment of combined virus infection especially in largemouth bass.

## 1. Introduction

Largemouth bass (*Micropterus salmoides*) is an economically important fish native to the southeastern United States that has been globally introduced to fish farms [1,2]. Largemouth bass is characterized by rapid growth, low temperature and disease resistance, and has rapidly progressed to a dominant aquaculture variety in China [3]. In 2021, the production of largemouth bass in China has reached 0.67 million tons, accounting for 99% of the world total harvest [4]. However, the continuous expansion of aquaculture, as well as overfeeding and environmental pollution, threatens this fish as well as other species of aquatic animals. In particular, mortality levels for farmed largemouth bass are high from viral and bacterial diseases that have no effective treatments [5,6]. There are numerous viruses that can infect farmed and wild largemouth bass and result in high mortality, including largemouth bass virus (LMBV) [7], largemouth bass ulcer syndrome virus (LBUSV) [8], micropterus salmoides rhabdovirus (MSRV) [5], birnavirus [9], infectious spleen and kidney necrosis virus (ISKNV) [10], piscine reovirus (PRV) [11], nervous necrosis virus (NNV) [12], spring viremia of carp virus (SVCV) [13] and viral hemorrhagic septicaemia virus (VHSV) [14]. The main virulent bacterial pathogens of this fish include *Nocardia seriolae* [15], *Edwardsiella piscicida* [16] and *Flavobacterium columnare* [17]. In addition, cultured fish are also susceptible to trypanosome [18] and *Myzobdella lugubris* [19] blood infections.

LMBV was first identified in 1995 at the Santee Cooper Reservoir (South Carolina) in the United States [20] and subsequently detected in New York, Pennsylvania, Texas and the Mississippi River, and has since spread to India and Asia. The virus was first identified in China in 2006 in a Guangdong fishery and has since spread across the entire country [21]. LMBV infections are associated with high mortality for farmed largemouth bass and the cumulative mortality is up to 100% [22]. MSRV was first identified in Europe in 2003 [23] and first emerged in a fishery in Zhongshan city, Guangdong, where 200,000 fish died in 20-hectare ponds [10]. The virus is now frequently detected in largemouth bass juveniles (1–4 cm), which are especially sensitive with a cumulative mortality up to 80% [10]. Anyway, these reports suggest that both LMBV and MSRV are the main pathogens for largemouth bass. Moreover, because of the complex water environment, infection with a single pathogen was no longer the main cause of death in aquatic animals while co-infections occurred more frequently. In 2018, multiple pathogens were found to co-infect largemouth bass with LMBV in various basins in Pennsylvania, causing large-scale mortality events [24]. Therefore, co-infections are more threatening and more difficult to treat than single infections and are more difficult to treat.

China is a primary largemouth bass breeding area and has suffered huge economic losses due to disease caused by LMBV, MSRV and various pathogenic bacteria [10,15,21]. However, viral co-infections have not been common in these fish populations. In this study, we simultaneously isolated and identified LMBV and hybrid snakehead rhabdovirus (HSHRV)-like virus from diseased largemouth bass and evaluated the virulence of this combination using artificial infection.

## 2. Materials and Methods

### 2.1. Ethics Statement

The research protocol and all the experimental procedures were approved by the Animal Care and Use Committee of Foshan University.

### 2.2. Sample Collection and Pathological Examination

Diseased largemouth bass were collected from a fish farm in Foshan, Guangdong Province, China and were enclosed in oxygenated plastic water bags for transport to the laboratory. After clinical signs were observed, the fish were anesthetized and complete necropsies were performed. Bacterial isolations were performed as described previously [25]. Portions of livers, spleens and kidneys were immersed in fixative containing 4% (*v*/*v*) paraformaldehyde (0.1 M phosphoric acid buffer, pH 7.0–7.5) for immunohistochemical examinations. The remaining pooled tissues were stored at −80 °C for virus detection and isolation. At the same time, tissues were also collected from the healthy fish and served as negative controls.

### 2.3. Cell Lines, Virus and Antibodies

*Micropterus salmoides* heart cells (MSH) were used for virus isolation [26] and were maintained at 28 °C in M199 medium (Gibco, Grand Island, NY, USA) containing 10% (*v*/*v*) fetal bovine serum (Trinity, Barcelona, Spain) and reduced to 4% for virus propagation. LMBV strain GD202006) and MSRV strain GD201707 were isolated and preserved in the laboratory as well as polyclonal antibodies mouse antiserum specific for LMBV and MSRV. The secondary antibody used for immunoassays were goat anti-mouse IgG horseradish peroxidase (HRP, Servicebio-Tek, Wuhan, China) and fluorescein isothiocyanate (FITC, Abcam, Cambridge, UK) conjugates.

### 2.4. Primers and Probes

Primer sets were synthesized by Sangon Biotech (Shanghai, China) for detection of LMBV, MSRV, ISKNV and NNV as well as amplification of the LMBV MCP and MSRV/MSRV-like virus G genes for sequence analysis. Viral load determinations in tissues utilized shorter qPCR primers for real time detection analysis (Table 1).

### 2.5. Virus Detection

Viral DNA and RNA were extracted using commercial total tissue DNA and RNA kits according to the instructions of the manufacturer (Omega, Guangzhou, China). RNA was reverse-transcribed using a commercial kit (Applied Biological Materials, Zhenjiang, China). The reaction system in a total volume of 20 µL consisted 2 μL template, 0.5 μL each primer (10 μM), 10 μL of 2× M5 HiPer plus Taq HiFi PCR Mix (with blue dye) (Mei5bio, Beijing, China) and 7 μL of nuclease- free water. Thermal cycler reactions were conducted with a heated lid and the following conditions: initial denaturation at 95 °C for 5 min, followed by 34 cycles at 94 °C for 25 s, 56 °C for 30 s and 72 °C for 60 s and a final extension at 72 °C for 5 min. Amplicons were visualized by 1% (*w*/*v*) agarose gel electrophoresis and sequenced at Sangon Biotech.

### 2.6. Immunohistochemistry

Immunohistochemistry (IHC) was performed as previously described with minor modifications [31,32,33]. In brief, fixed tissues were prepared as paraffin sections and hydrated with a graded series of xylene and ethanol and antigen recovery was performed by heating. The samples were then treated with 3% (*v*/*v*) hydrogen peroxide and sealed with serum. Mouse polyclonal antibodies against LMBV or MSRV were incubated overnight at 4 °C and goat anti-mouse IgG HRP conjugated were incubated at room temperature for 1 h. Positive reactions were visualized using diamino benzidine (DAB) and subsequently counterstained with hematoxylin for better visualization. These sections were then dehydrated, sealed and scanned by fluorescence microscope (Leica, Wetzlar, Germany).

### 2.7. Virus Isolation

Virus isolation was performed as previously described with minor modifications [34,35]. The pooled tissues were thawed at 4 °C and homogenized in Dulbecco’s PBS, subjected to three freeze–thaw cycles and differentially centrifuged at 3000, 8000 and 13,000× *g* for 10 min each time at 4 °C, using the supernatants of the preceding spins each time. The final supernatant was filtered (0.45-μm, Millipore, Burlington, MA, USA) and diluted 1:10 in cell culture medium. MSH monolayers were grown in Corning T-25 cell culture flasks (Corning, NY, USA) and inoculated with 2 mL of 10-fold serial dilutions of the filtered supernatants. Infected and non-infected cell cultures were incubated at 28 °C and examined daily for toxicity, contamination and viral cytopathic effects (CPE). The viruses were passaged three times and virus were harvested from cultures displaying >80% CPE and stored at −80 °C for subsequent research.

### 2.8. Morphology of Virus Particles

Infected MSH cells displaying clear CPE were collected and fixed at 4 °C with 2.5% (*v*/*v*) glutaraldehyde and then fixed with 1% osmium tetroxide in 0.2 M sodium cacodylate buffer for 1 h and dehydrated with a graded ethanol series. The specimens were then embedded in epoxy resin, sectioned and stained with 2% uranyl acetate-lead citrate. Virus particles were examined with a Hitachi HT-7650 transmission electron microscope (Tokyo, Japan).

### 2.9. Antigen Identification

Indirect immunofluorescence assays (IFA) were performed as described with appropriate modifications [36]. MSH cells cultured in 24-well plates were infected with viral supernatants and with LMBV-GD202006 and MSRV-GD201707 as positive controls; non-infected cells served as negative controls. The cells were incubated with maintenance medium for 4 d at 28 °C, then fixed in situ with −20 °C methanol for 30 min, followed by rinsing with PBS and then air-dried, added 0.5% (*v*/*v*) Triton X-100 (Sigma-Aldrich, St. Louis, MO, USA) and followed by incubation for 30 min at 25 °C. The cells were washed 3× with PBS containing 0.05% (*v*/*v*) Tween-20 (PBST, Sigma-Aldrich) and overlaid with PBST containing 5% (*w*/*v*) skimmed milk and incubated for 1 h at 25 °C. Polyclonal antibodies against LMBV or MSRV were then added at 1:200 and incubated at 37 °C for 1 h and were rinsed 3× with PBST. Goat anti-mouse FITC-labeled secondary antibodies (1:1000) were then added and incubated for 1 h at 37 °C and were rinsed with PBST. Propidium iodide (Thermo Fisher, Waltham, MA, USA) dye was then added to stain nuclei, and the cells were immediately observed using a DMI8 inverted fluorescence microscope (Leica).

### 2.10. Virus Plaque Assay

Plaque assays were carried out using MSH monolayers in 6-well plates, inoculated with 10-fold serial dilutions of virus supernatants and incubated for 1 h at 28 °C. Plates were then overlaid with medium containing 2% (*w*/*v*) low melting point agarose and followed by medium containing 0.002% (*v*/*v*) neutral red vital stain to construct a double layer of agarose and incubated for 3 d at 28 °C in a 5% CO_2_ incubator. Plaques of different sizes and shape were taken using sterile pipette tips and added to 500 µL M199 medium and the process was repeated until single plaque morphologies were obtained. The presence of virus was detected using PCR assays as described above.

### 2.11. Sequence and Evolutionary Analysis

The ORF sequences of rhabdovirus glycoprotein (G) and iridovirus major capsid protein (MCP) genes were amplified by PCR using primers as indicated (Table 1). The G and MCP genes were amplified using the respective PCR cycling conditions: 95 °C for 5 min, 94 °C for 30 s, 35 cycles of annealing (54 or 56 °C, resp.) for 30 s and extension at 72 °C for (60 or 80 s, resp.) and a final extension at 72 °C for 5 min. PCR products were purified and sequenced by Sangon Biotech. Alignments of predicted nucleic acid sequences were performed using BLAST (https://blast.ncbi.nlm.nih.gov/Blast.cgi, accessed 10 April 2022) and MEGA X (https://www.megasoftware.net/, download on 5 March 2022). Phylogenetic analyses were performed using MEGA X and determined analytical models for the gene sequences. MCP and G trees were constructed using the maximum-likelihood (ML) method and tests of phylogeny were done using 1000 replications bootstrap replicates.

### 2.12. Regression Infection Test

Healthy largemouth bass (20.0 ± 0.5 g) were kindly provided by a local fish farm in Foshan (Guangdong, China) and the liver, spleen, kidney of fish were selected and checked for the presence of LMBV, MSRV, ISKNV and NNV by PCR. A group of 180 healthy fish were then randomly divided into 4 groups (45 fish per group) that were intraperitoneally injected with 200 µL samples as follows: Negative control, sterile PBS; LMBV group, LMBV-FS2021 cell culture medium (5 × 10^5^ TCID_50_/mL); HSHRV group, HSHRV-MS2021 cell culture medium (1 × 10^6^ TCID_50_/mL); and co-infection group, 100 μL each of HSHRV-MS2021 (2 × 10^6^ TCID_50_/mL) and LMBV-FS2021 (1 × 10^6^ TCID_50_/mL). The mortality of experimental fish was observed and recorded every day for 16 d. Liver, spleen and kidney tissues of fish that died during the experiment and 5 surviving fish in groups at end of the experiment were collected for determination of viral load. In addition, samples were taken to re-isolate virus in MSH cells.

Virus-specific Taqman probes were used to determine viral loads using qPCR in the liver, spleen and kidney of fish in each group to confirm that the fish died of LMBV or/and HSHRV infections. The commercial One Step Reverse Transcription Kit and Pro Taq HS Premix Probe qPCR Kit (AGbio, Changsha, China) were used for qPCR. qPCR was performed as previously described with appropriate modifications [29,30]. In brief, qPCR reaction system was carried out in a final volume of 20 μL, which contained 10 μL 2× Pro Taq HS Probe Premix, 0.4 μL each primer (10 μM), 0.4 μL probe (0.2 μM), 0.4 μL ROX Reference Dye (4 μM), 2 μL template (2 μL of 50 ng/μL) and 6.4 μL water. The PCR procedure was performed using an ABI7500 instrument (Applied Biosystems, Foster City, CA, USA) as follows: 95 °C for 5 min, followed by 40 cycles of 95 °C/5 s and 60 °C/30 s. All samples were run in duplicate. A standard curve was generated using 10-fold dilutions of purified standard virus genome (range 10^0–^10^8^ copies/μL) and were linear within this range (LMBV: Y = −3.4719x + 34.4208, HSHRV: Y = −3.163x + 45.216).

### 2.13. Statistical Analyses

The statistical differences among experimental groups were determined by analysis of variance using GraphPad Prism software V8.0.2.263 (San Diego, CA, USA). A *p* value of <0.05 was considered statistically significant.

## 3. Results

### 3.1. Clinical Characteristics

The primary clinical symptoms of the diseased fish we procured from the local fish farm were ulcer lesions on body surface and different-sized foci and muscle necrosis as well as mild bleeding appeared on the body surface (Figure 1A). In addition, livers were swollen and displayed white nodules and had lost color while the intestines were adherent and swollen (Figure 1B). Bleeding and necrosis were also apparent in the gills (Figure 1C).

### 3.2. Bacterial Isolation and Virus Detection

We attempted but could not isolate bacteria from tissue samples of the diseased fish and ruled out the possibility of bacterial infections. In contrast, PCR detection using DNA and RNA isolated from homogenized tissues showed that only primers for LMBV and MSRV amplified specific target bands consistent with the expected size, ~430 bp and ~349 bp, respectively (Figure 2), indicating the presence of LMBV and MSRV while NNV and ISKNV were absent. Sequence alignments indicated that LMBV-primer derived amplicons were 99% identical to LMBV and MSRV-primer amplicons were 98.5% identical to HSHRV and 90.8% identical to MSRV (data no shown). These results indicated that the diseased fish were co-infected with LMBV and the HSHRV-like virus.

### 3.3. Immunohistochemistry (IHC)

Anti-LMBV and anti-MSRV polyclonal antibodies were used to detect specific viruses in tissues of diseased fish by using IHC. We found the presence of immunogenic signals for LMBV and MSRV (or similar viruses) in livers, spleens and kidneys of the diseased fish, although the liver tissues were only weakly positive (Figure 3). In general, LMBV immunogenic signals (Figure 3D2,E2,F2) were significantly greater than MSRV (Figure 3A2,B2,C2) in all three tissues. Additionally, the kidneys (Figure 3C,F2) were more immunoreactive than the livers (Figure 3A2,D2) and spleens (Figure 3B2,E2).

### 3.4. Virus Isolation

We used MSH cells to isolate virus from pooled liver, kidney and spleen homogenates and CPE were apparent as early as 1 day post-infection (dpi) (Figure 4B). The cells displayed signs of shrinkage and death at 5 dpi (Figure 4C) and the monolayers were finally destroyed. We also generated PCR amplicons using LMBV-specific primers and MSRV-specific primers, and the results were consistent with those shown in Figure 2 (data not shown). Together, these data indicated that the diseased fish had been co-infected with LMBV and MSRV (or similar viruses).

### 3.5. Morphology of Virus Particles

The virions isolated from virus samples in infected MSH cells were structurally examined using TEM. We found two different virion types: a small number of spindle and bullet-shaped virions that were 100–140 nm in length and 50–100 nm in diameter were appeared in the cytoplasm and nuclei (Figure 5A). The second type were numerous particles with typical iridovirus characteristics of approximate 140 nm diameters [37] that were scattered throughout the cytoplasm (Figure 5B).

### 3.6. Antigen Identification

We also examined infected MSH cells for reactivity to polyclonal antibodies that were previously developed against LMBV and MSRV. Both LMBV (Figure 6A) and MSRV (Figure 6E) fluorescent signals were apparent in cells infected with viral supernatants. Cells infected with the positive control viruses LMBV-GD202006 (Figure 6B) and MSRV-GD201707 (Figure 6F) also displayed significant fluorescence signals for these antibodies. The respective antibodies did not cross-react with cells infected with non-cognate positive control virus (Figure 6C,G) and uninfected negative control cells lacked any fluorescent signal (Figure 6D,H).

### 3.7. Virus Plaque Assays

The presence of co-infection with LMBV and HSHRV-like viruses were also apparent in cell cultures of infected with virus supernatants. Viral plaques of different sizes and shapes were apparent at 4 dpi, and following 3 rounds of plaque purification, we identified LMBV and HSHRV-like virus using PCR (data not shown).

### 3.8. Sequence Analysis of Conserved Regions of Viral Genome

The purified virions were then used to amplify the complete ORFs sequences of the highly conserved MCP protein (iridovirus) and protein G (rhabdovirus) that were 1391 and 1526 bp, and the sequences were deposited in GenBank under accession numbers ON418985 and ON418986, respectively. A comparative sequence analysis revealed that the MCP gene of the newly isolated iridovirus shared 100.00, 99.87 and 99.01% identity with the LMBV isolates Santee-Cooper ranavirus BG/TH/CU3, LBUSV-EPC060608-08 and LMBV-Pine-14-204, respectively. The newly isolated iridovirus was clustered with Santee-Cooper ranavirus, LMBV and LBUSV into the same genus, but was clustered with Santee-Cooper ranavirus (actually LMBV) into the same group, indicating that this iridovirus is closely related to LMBV, so it is temporarily assigned the name LMBV-FS2021 (Figure 7). The G gene of the newly isolated rhabdovirus revealed 98.16 and 90.94% identity with HSHRV-C1207 and MSRV-YH01, respectively. The newly isolated rhabdovirus was clustered with HSHRV, SCRV and MSRV into the same genus, but was clustered with HSHRV into the same group, indicating that this rhabdovirus is closely related to HSHRV, so it is temporarily assigned the name HSHRV-MS2021 (Figure 8).

### 3.9. Regression Infection Experiments

Since our results indicated that the original diseased fish were co-infected, we examined the pathogenicity of the two new viruses alone and in combination. Cell culture supernatants from LMBV-FS2021 and HSHRV-MS2021-infected MSH cells were injected into healthy largemouth bass both alone and in combination. The LMBV-FS2021 and co-infection group began to die at 1 dpi and the fastigium death occurred at 3 dpi. However, the HSHRV-MS2021 group began to die at 7 dpi and a total of five fish died during the whole test. The cumulative mortality rate in HSHRV-MS2021 and LMBV-FS2021 infected groups were 11.1% and 86.7%, respectively (Figure 9A). The cumulative mortality of the co-infection group was 93.3% and significantly higher than those infected with either virus alone (*p* < 0.05) (Figure 9A). In order to evaluate the viral load in fish that succumbed and survived infection, five fish from each group were randomly selected after death or survival at 16 dpi and pooled liver, spleen and kidney homogenates were assayed using qPCR. We found that the average viral loads of LMBV-FS2021-infected and HSHRV-MS2021-infected fish were 2.75 × 10^5^ and 3.21 × 10^3^ copies/μL, respectively. In contrast, the viral loads of LMBV-FS2021 and HSHRV-MS2021 for the co-infections were 3.08 × 10^5^ and 7.41 × 10^2^ copies/μL, respectively (Figure 9B). In addition, qPCR results using tissues from all the dead fish were virus-positive, while uninfected fish were all virus-negative, and the LMBV-FS2021 and/or HSHRV-MS2021 were isolated again from virus-infected fish.

## 4. Discussions

Iridoviridae members are non-enveloped dsDNA viruses with icosahedral symmetry [37] and there are currently two subfamilies *Alphairidovirinae* and *Betairidovirinae* and 7 genera; *Lymphocystivirus*, *Megalocystivirus*, *Ranavirus*, *Chloriridovirus*, *Daphniairidovirus*, *Decapodiridovirus* and *Iridovirus* (https://talk.ictvonline.org/taxonomy/, accessed on 30 May 2022). LMBV, epizootic haematopoietic necrosis virus (EHNV), ECV, Singapore grouper iridovirus (SGIV) and ISKNV are iridoviruses that infect freshwater fish. Taxonomically, EHNV, ECV, LMBV and SGIV belong to the genus *Ranavirus* within the *Alphairidovirinae*.

Rhabdoviruses are enveloped ssRNA with bullet-shaped virions belonging to the *rhabdoviridae* [38]. Currently, members of the *rhabdoviridae* are divided into three subfamilies, *Alpharhabdovirinae*, *Betarhabdovirinae* and *Gammarhabdovirinae*, that together contain 45 genera (https://talk.ictvonline.org/taxonomy/, accessed on 30 May 2022). MSRV, SCRV, HSHRV, SVCV, VHSV and IHNV (infectious hematopoietic necrosis virus) are freshwater fish-infecting rhabdoviruses. Taxonomically, SCRV, MSRV and HSHRV are groups in the genus *Siniperhavirus* within the *Alpharhabdovirinae* [39]. Previous studies had indicated that both MSRV and SCRV can infect largemouth bass [10,40] and some MSRV shared 100.00% identity with SCRV genomes (https://blast.ncbi.nlm.nih.gov/Blast.cgi, accessed on 23 April 2022, data not shown). This suggests that MSRV and SCRV may be the same virus from different hosts. Meanwhile, HSHRV and SCRV/MSRV possess highly similar genomes sequence.

The primary clinical symptoms of largemouth bass infections with LMBV are pale nodules on the surface of viscera, necrosis of spleen, skin and muscles. These infections also result in leaving white necrotic areas that are uplifted in the centers and surrounded by bleeding [22]. The moribund largemouth bass collected in this study displayed these typical clinical symptoms including white liver nodules, organ enlargement and muscle necrosis and bleeding. However, the diseased fish also showed some clinical symptoms that are inconsistent with LMBV infection [1,40]. These led us to speculate that the diseased largemouth bass infected with LMBV may also be infected with other pathogens. Therefore, bacterial isolation and PCR detection of viruses common to largemouth bass were performed. No bacteria were isolated from the visceral tissues of the diseased largemouth bass, ruling out the possibility of bacterial disease. PCR amplification results were positive using LMBV and MSRV primers, indicating that the diseased fish were infected with both LMBV and MSRV or like-viruses, and subsequent studies confirmed this. Further IHC results confirmed that the diseased fish were infected with LMBV and MSRV (actually an HSHRV-like virus). We found immune reactivity in liver, spleen and kidney, and the kidney displayed the most robust response, suggesting the latter as the primary viral target organ, as is consistent with previous reports [41]. Moreover, LMBV immunoreactivity was higher than that of MSRV and indicated that LMBV was most likely the primary pathogen of these diseased largemouth bass.

We were able to plaque-purify both viruses, and the iridovirus was a new isolate of LMBV, temporarily named LMBV-FS2021, and a rhabdovirus isolate was most similar to the hybrid snakehead rhabdovirus (HSHRV), temporarily named HSHRV-MS2021. HSHRV was first isolated from hybrid snakehead in 2014 [42] and the clinical symptoms of infected fish include the appearance of petechiae on the tail fin, hepatomegaly and serious hemorrhages on the surface of the swim bladder that often leads to high morbidity. HSHRV is an important pathogen in hybrid snakehead and the current study is the first to identify this as a pathogen in largemouth bass, although it was not as pathogenic as LMBV-FS2021 in our study.

Our results confirmed that the diseased largemouth bass from the Foshan farm were co-infected with LMBV and HSHRV. To our knowledge, this is the first reported of largemouth bass being infected with both viruses. Co-infection of multiple pathogens is a very common phenomenon in aquaculture with combinations of parasites, bacteria or viruses. SCRV and ISKNV co-infections of siniperca chuatsi [43], infectious pancreatic necrosis virus (IPNV) and IHNV co-infections of *Oncorhynchus mykiss* [44] as well as infectious hypodermal and hematopoietic necrosis virus (IHHNV) and white spot syndrome virus (WSSV) in wild crustaceans [45] have been reported. Together with our data that the co-infection of viruses increases virulence and poses a huge challenge to the health protection of aquatic animals, our results indicated that co-infection increased fish mortality even though viral loads differed by more than 100-fold and was a significant (*p* < 0.05) difference between single and co-infection. A possible explanation for these results is that LMBV replication is promoted following co-infection because we found the viral loads for LMBV in the co-infection group was significantly higher than that with LMBV alone. Previous studies have also shown synergistic actions of the two viruses significantly increase morbidity and mortality in infected fish [43,44]. The specific influence and mechanism of the interaction between LMBV-FS2021 and HSHRV-MS2021 awaits further study.

## 5. Conclusions

In this study, the virus LMBV-FS2021 and HSHRV-MS2021 were simultaneously isolated and identified from diseased largemouth bass by immunohistochemistry, cell culture, electron microscopy, indirect immunofluorescent assay and conserved gene sequence analysis. In addition, regression infection experiments indicated that the LMBV-FS2021 was highly virulent to largemouth bass while HSHRV-MS2021 was weakly virulent and co-infection increased mortality. This is the first report of co-infection of LMBV and HSHRV in largemouth bass and provides new data and information for prevention and control of viral diseases of this fish.

## Figures and Tables

**Figure 1 viruses-14-01643-f001:**
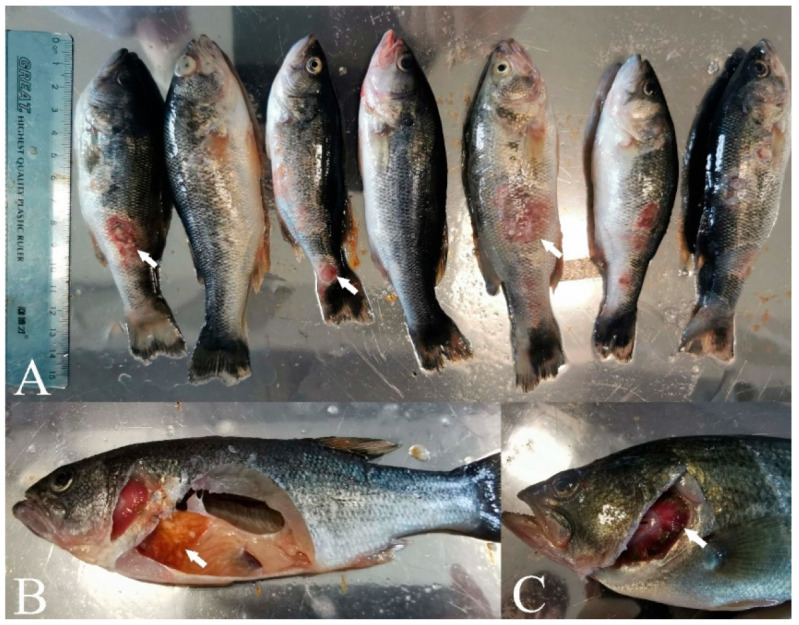
Clinical characteristics of diseased largemouth bass. (**A**) Ulcer lesions on the body surface (white arrows), (**B**) lesions in the viscera (white arrows), and (**C**) bleeding and necrosis lesions on the gills (white arrows).

**Figure 2 viruses-14-01643-f002:**
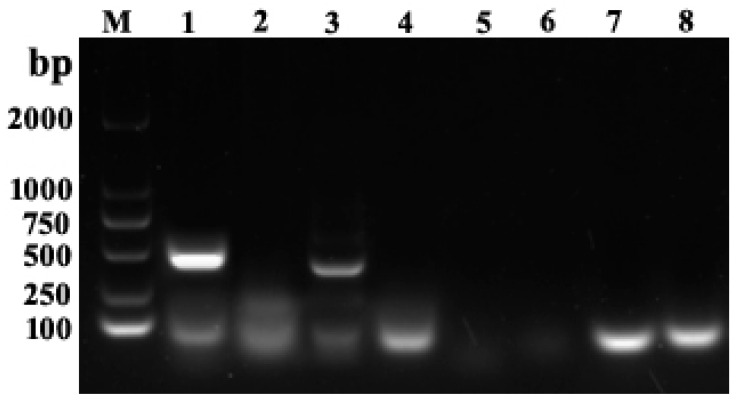
PCR amplicons used for virus detection from tissue homogenates of diseased largemouth bass. (M) A 2000 bp DNA marker, (Lane 1) LMBV detection using the tissue of diseased fish as template, (Lane 2) negative control with water as template, (Lane 3) MSRV detection using the tissue of diseased fish as template, (Lane 4) negative control with water as template, (Lane 5) NNV detection using the tissue of diseased fish as template, (Lane 6) negative control with water as template, (Lane 7) ISKNV detection using the tissue of diseased fish as template, and (Lane 8) negative control with water as template.

**Figure 3 viruses-14-01643-f003:**
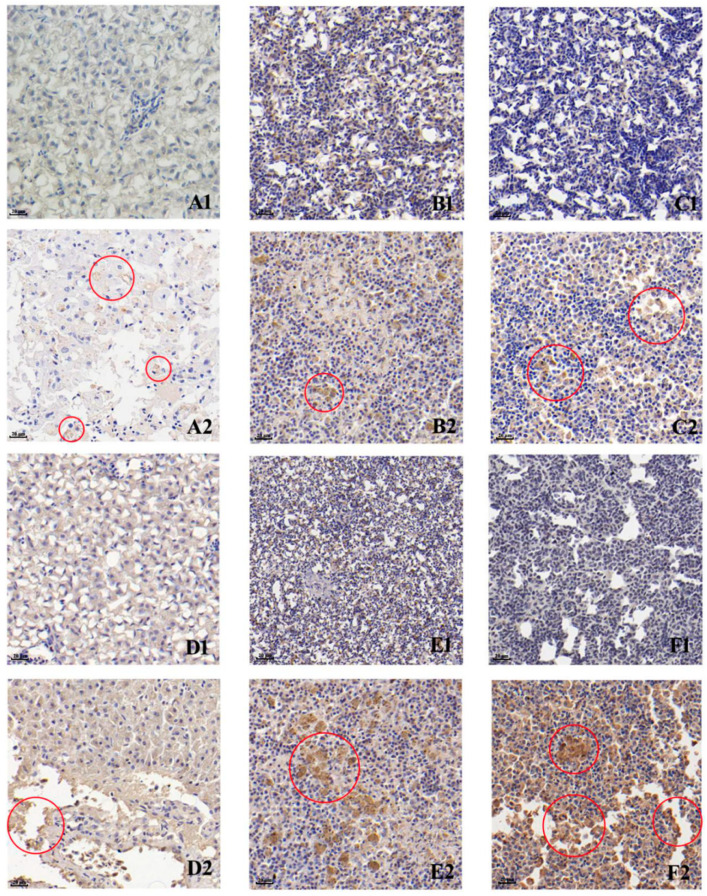
Immunohistochemistry detection of LMBV and HSHRV-like virus in naturally virus-infected largemouth bass suing specific polyclonal antisera. Anti-MSRV staining of livers of (**A1**) healthy and (**A2**) diseased fish; spleens of (**B1**) healthy and (**B2**) diseased fish; and kidneys of (**C1**) healthy and (**C2**) diseased fish. Anti-LMBV staining of (**D1**) healthy and (**D2**) diseased fish; spleens of (**E1**) healthy and (**E2**) diseased fish; and kidneys of (**F1**) healthy and (**F2**) diseased fish. Scale bars, 20 μm.

**Figure 4 viruses-14-01643-f004:**
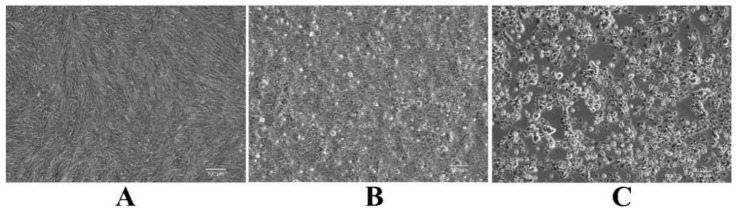
The CPE of *Micropterus salmoides* heart (MSH) cells infected with new isolated virus in MSH cell monolayers. (**A**) Uninfected MSH cells, (**B**) MSH cells at 1 day post-infection (dpi) with the new isolated virus, and (**C**) MSH cells at 5 dpi with the new isolated virus. Bar = 100 µm.

**Figure 5 viruses-14-01643-f005:**
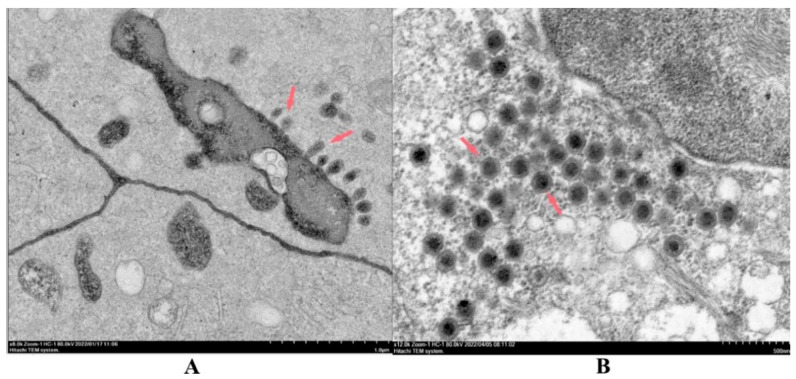
TEM analysis of infected MSH cells. All samples were prepared from the same batch and were observed using TEM, 80.0 kV. (**A**) Rhabdovirus-like viral particles (red arrows), and (**B**) iridovirus-like viral particles (red arrows).

**Figure 6 viruses-14-01643-f006:**
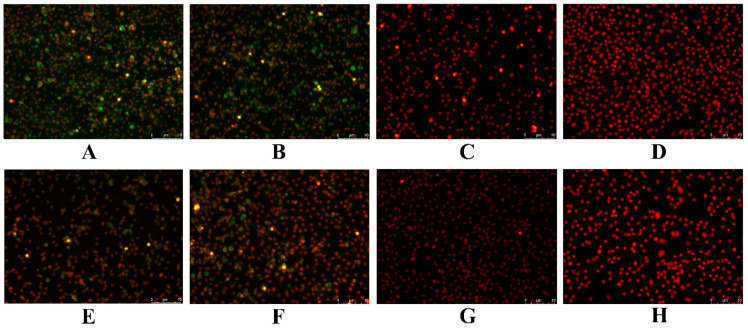
Indirect immunofluorescence assays for infected MSH cells. (**A**) New isolated virus infection/anti-LMBV Ab, (**B**) LMBV-GD202006 infection/anti-LMBV Ab, (**C**) LMBV-GD202006 infection/anti-MSRV Ab, (**D**) uninfected cells/anti-LMBV Ab, (**E**) New isolated virus infection/anti-MSRV Ab, (**F**) MSRV-GD201707 infection/anti-MSRV Ab, (**G**) MSRV-GD201707 infection/anti-LMBV Ab, and (**H**) uninfected cells/anti-MSRV Ab. The nuclei were stained red using propidium iodide (PI) solution. Bar = 10 μm.

**Figure 7 viruses-14-01643-f007:**
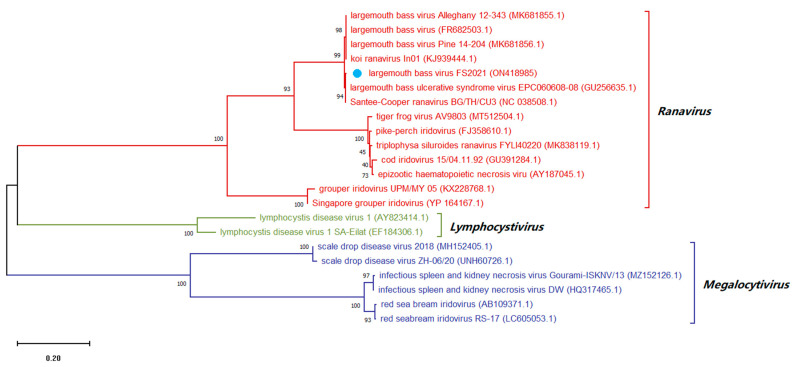
Maximum-likelihood tree of the major capsid protein (MCP protein) gene sequence of LMBV-FS2021. LMBV-FS2021 is designated by the blue dot. GenBank accession numbers are shown in brackets. The evolutionary history was inferred by using the Maximum Likelihood method and Kimura 2-parameter model. The tree with the highest log likelihood (−10532.26) is shown. The percentage of trees in which the associated taxa clustered together is shown next to the branches. Initial tree(s) for the heuristic search were obtained automatically by applying Neighbor-Join and BioNJ algorithms to a matrix of pairwise distances estimated using the Maximum Composite Likelihood (MCL) approach, and then selecting the topology with superior log likelihood value. The tree is drawn to scale, with branch lengths measured in the number of substitutions per site. This analysis involved 22 nucleotide sequences. There were a total of 1819 positions in the final dataset. Evolutionary analyses were conducted in MEGA X.

**Figure 8 viruses-14-01643-f008:**
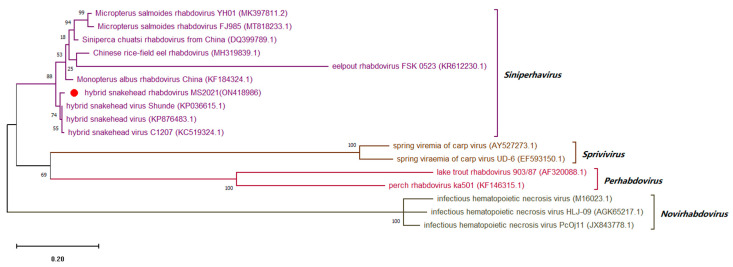
Maximum-likelihood tree of the glycoprotein (G protein) gene sequence of HSHRV-MS2021. HSHRV-MS2021 is designated by the red dot. GenBank accession numbers are shown in brackets. The evolutionary history was inferred by using the Maximum Likelihood method and General Time Reversible model. The tree with the highest log likelihood (−14131.39) is shown. The percentage of trees in which the associated taxa are clustered together is shown next to the branches. Initial tree(s) for the heuristic search were obtained automatically by applying Neighbor-Join and BioNJ algorithms to a matrix of pairwise distances estimated using the Maximum Composite Likelihood (MCL) approach, and then selecting the topology with superior log likelihood value. The tree is drawn to scale, with branch lengths measured in the number of substitutions per site. This analysis involved 17 nucleotide sequences. There were a total of 2347 positions in the final dataset. Evolutionary analyses were conducted in MEGA X.

**Figure 9 viruses-14-01643-f009:**
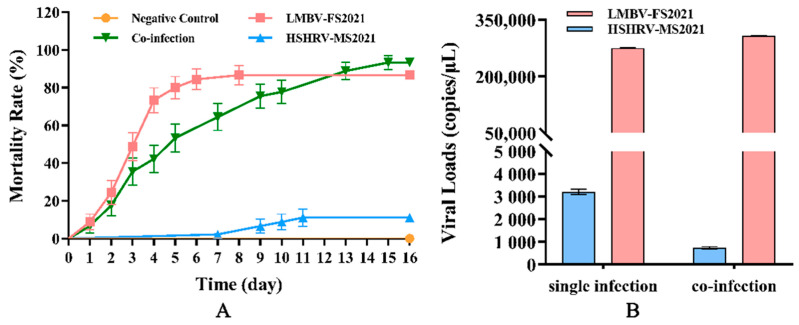
Cumulative mortality rate and viral loads of largemouth bass after challenge with viruses. (**A**) Cumulative mortality curves of largemouth bass in each group at 16 dpi, and (**B**) virus loads in largemouth bass at 16 dpi. The graph was produced by GraphPad Prism V8.0.2.263.

**Table 1 viruses-14-01643-t001:** Primers and probes used in this study.

Name	Sequence (5′-3′)	Product Size (bp)	References/Purpose
LMBV-DT-F	GCTGGCGGCCAACCAGTTTAAC	430	This study Virus detection
LMBV-DT-R	GGCCACGATTGGCTTGACTTCT		
MSRV-DT-F	GGGCTGGATGATAGACGATTG	349	This study Virus detection
MSRV-DT-R	TGGCGGAGGTGCTTGATATGG		
ISKNV-DT-F	GGTTCATCGACATCTCCGCG	431	[27] Virus detection
ISKNV-DT-R	AGGTCGCTGCGCATGCCAATC		
NNV-P0	CGAGTCAACACGGGTGAAGACAG	326	[28] Virus detection
NNV-P1	ACCGCTCCCATCATGACACAA		
NNV-P2	AACAGGCAGCAGAATTTGACG		
MCP-F	ACCAACATTTCTATCGCTTAT	1456	This study Sequencing
MCP-R	TGCGATATGGAAACGTAGTAA		
G1-F	ATTAATCAATGGTGTTGGTGG	946	This study Sequencing
G1-R	CCACCAACACCATTGATTAAT		
G2-F	CATATCCGAATTGCGAAGAGC	637	This study Sequencing
G2-R	TTCTTGAGAATAATCCATGAT		
LMBV-qPCR-F	GGCCACCACCTCTACTCTTAC	120	[29] Virus loads determination
LMBV-qPCR-R	GGCAGACAGAGACACGTTGA		
LMBV-probe	FAM-CTTCAGGGTCTACCAATTTCGGTC-TAMRA		
MSRV-qPCR-F	GACATGTTCTTCTACAGATTCAAC	140	[30] Virus loads determination
MSRV-qPCR-R	CAATCCAGCACTCCACTG		
MSRV-probe	FAM-AGGTTCAAAGACTGTGCAGCTCTGT-TAMRA		

## Data Availability

All of the materials and data that were used or generated in this study are described and available in the manuscript.

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
