# Peer review of "Simultaneous Isolation and Identification of Largemouth Bass Virus and Rhabdovirus from Moribund Largemouth Bass (Micropterus salmoides)"

_viruses, 2022, doi:10.3390/v14081643_

Round 1

Reviewer 1 Report

1. The title should be carefully checked!!

2. The parts of this manuscript including abstract, introduction, results, discussion, etc. all need to be improved carefully! There are lots of puzzled sentences!!

3. The information about PCR amplification should be detailed added. And it is confused for the marked “bp” in Fig.2. The contents in line 230-233 should be described in the section 3.8. Besides, the contents of PCR amplification should be presented together with sequence analysis.

4. Line 345 showed the significant difference of P<0.01, there was no mentioned in the statistical analyses.

5. Captions of Fig.4, 5, 7, 8 and 9 etc. should be modified. The statements are not clear.

6. “samples were taken to re-isolate virus in MSH cells” was mentioned in the method, but the results cannot find.

7. What the “ The ultimate mortality rate”?

8.  In line 353, it is wrong indication for figure 9, the figure for “ viral load in fish” should be added.

Author Response

Thank you very much for your comments.

Please find the following response to the comments:

  1. The title should be carefully checked!!

Response: Thank you for pointing this out. We have corrected the title.

  1. The parts of this manuscript including abstract, introduction, results, discussion, etc. all need to be improved carefully! There are lots of puzzled sentences!!

Response: Thank you for your kind suggestion and comments. we have checked and revised the whole MS carefully.

  1. The information about PCR amplification should be detailed added. And it is confused for the marked “bp” in Fig.2. The contents in line 230-233 should be described in the section 3.8. Besides, the contents of PCR amplification should be presented together with sequence analysis.

Response: Thank you for your kind suggestions. We have supplemented the information of PCR amplification in the update MS, “...RNA isolated from homogenized tissues showed that only primers for LMBV and SMRV amplified specific target bands consistent with the expected size, ~430bp and ~349bp, respectively (Figure 2). Indicating the presence of LMBV and MSRV while NNV and ISKNV were absent....” .

   “bp” in Fig.2 is the units of the DNA marker band, we marked “bp” in Fig.2 can be clearly shown that the corresponding size of each bands are 2000bp,1000bp 750bp..... other than 2000kb, 1000kb 750kb....

We apologize for any confusion this may have caused. Virus detection in section 3.2 and sequence analysis of conserved regions of viral genome in section 3.8 are two different things. The primers used for virus detection in section 3.2 only amplify a small portion of the MCP or G gene sequence, the purpose of sequencing is to determine the authenticity of the PCR products and to identify the virus. While the entire ORF of MCP or G gene of the two newly isolated viruses were amplified and sequenced in section 8 for sequence alignment and genetic evolution analysis.

  1. Line 345 showed the significant difference of P<0.01, there was no mentioned in the statistical analyses. Captions of Fig.4, 5, 7, 8 and 9should be modified. The statements are not clear.

Response: Thank you for your comments. It should be P<0.05, we have revised it in the updated MS. We have also modified the captions for all the figures to make them clearer.

  1. Captions of Fig.4, 5, 7, 8 and 9should be modified. The statements are not clear.

Response: Thank you for your kind suggestion. We have modified these captions in the updated MS.

  1. “samples were taken to re-isolate virus in MSH cells” was mentioned in the method, but the results cannot find.

Response: Thank you for pointing this out. we are sorry that we didn't mention it in the results section, we have added “LMBV-FS2021 and/or HSHRV-MS2021 were isolated again from virus infected fish ” Line 367-368 in the section of “3.9. Regression infection experiments ” in updated MS.

  1. What the “ The ultimate mortality rate”?

Response: Thank you for your kind question. It should be “The cumulative mortality” , we have revised it in the updated MS.

  1. In line 353, it is wrong indication for figure 9, the figure for “ viral load in fish” should be added.

Response: Thank you for your kind reminder. We have modified the indication in updated MS, and added the figure 9B for viral load in fish.

Reviewer 2 Report

Jin et al. isolated LMBV and MSRV simultaneously from moribund largemouth bass and evaluated the characterization of these two viruses. It is interesting and useful for studies of viral co-infection. I have some concerns to be addressed.

1.     the title can be changed to “Simultaneous Isolation and Identification of Largemouth Bass Virus and a novel Rhabdovirus from Moribund Largemouth Bass (Micropterus Salmoides)”.

2.     In figure 1, there are at least seven diseased fish, are the LMBV and MSRV simultaneously isolated from all the seven fish or just one or two fish. 

3. the legend of figure2, the authors need to present each lane clearly. lane1: LMBV detection using the tissue of diseased fish as template, lane2: negative control is what? the template is water or healthy tissue.

4. In 3.5, the authors purified the two viruses using plaque, is there any differences on the plaque size caused by the two viruses.

5. In figure 6, (A) and (E) what is the wild type virus? What is used for the red signal. 

6. In 3.9, what is the dose of the two viruses injected into largemouth bass. Is the LD50 of the two viruses evaluated in largemouth bass? 

Author Response

Thank you very much for your comments.

Please find the following response to the comments: 

  1. the title can be changed to “Simultaneous Isolation and Identification of Largemouth Bass Virus and a novel Rhabdovirus from Moribund Largemouth Bass (Micropterus Salmoides)”.

Response: Thank you for your nice suggestion. We have revised it according to your opinion.

  1. In figure 1, there are at least seven diseased fish, are the LMBV and MSRV simultaneously isolated from all the seven fish or just one or two fish. 

Response: Nice question. In fact, we collected more than 15 diseased fish, and both viruses were detected in eight fish, then we mixed the tissue samples of the two viruses detected together to isolate the viruses. Some diseased fish only detect LMBV and are not covered in this paper.

  1. the legend of figure2, the authors need to present each lane clearly. lane1: LMBV detection using the tissue of diseased fish as template, lane2: negative control is what? the template is water or healthy tissue.

Response: Thank you for your kind suggestion. We have revised them according to your opinion. Lane 2: negative control with water as a template.

  1. In 3.5, the authors purified the two viruses using plaque, is there any differences on the plaque size caused by the two viruses.

Response: Nice question. There is no absolute differences on the plaque size caused by the two viruses, but we found that the most of larger plaques were LMBV, and the small plaques are either LMBV or HSHRV or both. We randomly selected some plaques of different sizes and shapes for identification  and for the next round of plaque tests until single type of virus were obtained.

  1. In figure 6, (A) and (E) what is the wild type virus? What is used for the red signal. 

Response: Thank you for your kind question. We apologize for confusing you with our description, it should be the new isolated viruses, we have revised it in the updated MS.

The nuclei can be stained red using propidium iodide (PI) solution, which facilitates clear observation of the target fluorescence signal (green fluorescence signal in this study).

  1. In 3.9, what is the dose of the two viruses injected into largemouth bass. Is the LD50of the two viruses evaluated in largemouth bass? 

Response: Thank you for your kind question. In line 194-197, we have stated the titers and doses of virus culture medium used in the infection experiments.

Great questions. The injection doses of two viruses are described in line 195-197 of the manuscript. We only evaluated the LD50 of LMBV-FS2021 (1.68×105 TCID50/ mL for largemouth bass at ~ 20 g ). We used 3-fold LD50 as the injection dose to carry out the regression infection assay. While the fatality rate of HSHRV-MS021 to largemouth bass was too low and its LD50 could not be determined.

Reviewer 3 Report

The authors reported that a largemouth bass virus and a novel rhabdovirus HSHRV were simultaneously isolated and identified from the moribund largemouth bass using a combination of immunohistochemistry, cell culture, electron microscopy and conserved gene sequence analysis.  Regression infection experiments indicated fish mortalities for LMBV and HSHRV of 86.7 and 11.1 %, respectively. while co-infection of these two virus resulted in 93.3 % mortality that was significantly different from the single infections. The results obtained from this study will be beneficial  for the prevention and treatment of combined virus infection especially in largemouth bass.I recommend publication of the manuscript after addressing of the comments shown below:

1. Line 3, In title, ... Moribund the... should be .... the Moribund... ;

2. Line 35, 100=fold should be 100-fold;

3. There is extra space in many places of the MS, such as Line 67, Line 105, Line 148, Line 364, et al. Please check them in the whole MS.

4. Line 76-78, please add references to this point.

5. Line 97, M, salmoides should be use full name.

6. Line 103, These polyclonals were made in mice? Please check.

7. Line 103, as were.should be as well..

8. Line 104, delete tissues

9. Line 185, if you are using the R package for analysis, you will need to refer to the original literature of R package.

10. Line 204, please specify the complete qPCR procedure.

11. The line spacing of references needs to be adjusted.

Author Response

Thank you very much for your comments.

Please find the following response to the comments: 

  1. Line 3, In title, ‘... Moribund the...’ should be ‘.... the Moribund...’ ;

Response: Thank you for pointing our mistake. We have revised it in updated MS.  

  1. Line 35, 100=fold should be 100-fold;

Response: Thank you for pointing our mistake. We have revised it in updated MS.

  1. There is extra space in many places of the MS, such as Line 67, Line 105, Line 148, Line 364, et al. Please check them in the whole MS.

Response: We have checked the whole MS and deleted these spaces.

  1. Line 76-78, please add references to this point.

Response: Thank you for your kind reminder. We have added references for it.

  1. Line 97, M, salmoidesshould be use full name.

Response: Thank you for your suggestion. We changed the name of the cell line to “Micropterus Salmoides Heart cells (MSH)”.

  1. Line 103, These polyclonals were made in mice? Please check.

Response: Thank you for your kind question. We re-examined the polyclonal antibody and confirmed that it was made from mice.

  1. Line 103, ‘as were.’ should be ‘as well..’

Response: We have revised it in updated MS.

  1. Line 104, delete tissues

Response: We have deleted it.

  1. Line 185, if you are using the R package for analysis, you will need to refer to the original literature of R package.

Response: Thank you for your kind suggestion. In order to show the evolutionary analysis results more clearly, we have switched to using MEGA X software to make evolutionary trees in updated MS.

  1. Line 204, please specify the complete qPCR procedure.

Response: Thank you for your kind suggestion. We have added the complete qPCR procedure in updated MS.

  1. The line spacing of references needs to be adjusted.

Response: Thank you for pointing our mistake. We have revised them in updated MS.

Reviewer 4 Report

This paper describes the isolation of two co-infecting viruses from moribund largemouth bass commercially farmed in China. Clinical signs in the diseased fish are described. Immunohistochemistry, electron microscopy and amplified gene sequence analysis indicate that the infectious agents were an iridovirus (largemouth bass virus; LMBV) and a rhabdovirus (hybrid snakehead rhabdovirus; HSHRV). The viruses were isolated and plaque-purified. Experimental infections and mortality rates are reported, both as single and dual infections. 

This is a relatively well written paper and the reported results are generally useful. However, there are several issues that the authors should address.

1.     Phylogenetic inferences are poorly constructed and presented. Neighbour-joining trees provide a very crude inference of evolutionary relationships; maximum-likelihood trees are easily constructed with modern software and are preferred for the type of analysis conducted here. Furthermore, the sequence alignment algorithm and lengths of aligned sequences are not disclosed, no bootstrap values are shown, the branch length scales are not defined, and branch lengths are not proportional to evolutionary distance.

2.     For the rhabdovirus tree, a larger set of viruses should be used to place the isolate and the relationships to other viruses in a suitable context. This is particularly important as the authors refer to the rhabdovirus as a “novel” virus. However, all of the sequences presented in the tree are from isolates assigned to the same rhabdovirus species (Siniperhavirus chuatsi). Indeed, the authors report the G protein sequence of the rhabdovirus isolate to share 98.16% amino acid sequence identity with the G protein of HSHRV strain C1207 which is well within the range of identify for isolates of the same virus. Furthermore, the authors do not include Chinese rice-field eel rhabdovirus (MH319839) in their analyses and this is also assigned to the same rhabdovirus species. As a minimum effort, the authors should include all of these isolates in their tree and use, as an outgroup, a virus assigned to another closely related rhabdovirus species. Eelpout rhabdovirus (KR612230; species Siniperhavirus zoarces) would be a suitable outgroup.  A larger tree placing the new isolate in the context of other alpharhaboviruses infecting fish and marine mammals would be preferred.

3.     Reference to rhabdovirus MS2021 as a “novel virus” should be removed throughout. It is simply another isolate of a previously known and described virus. 

4.     The use of virus nomenclature and taxonomic classification throughout does not conform to conventions established by the International Committee on Taxonomy of Viruses (ICTV).  Virus names should not be capitalised unless they include proper nouns. This applies both in the text and in the figure legends. Virus species names should be italicised and should comply with the most recent release of the ICTV master species list (MSL); see https://talk.ictvonline.org/taxonomy.  The authors will also find the following reference to the revised taxonomy of rhabdoviruses infecting fish and marine mammals useful: https://doi.org/10.3390/ani12111363.

Author Response

Thank you very much for your comments.

Please find the following response to the comments:

  1. Phylogenetic inferences are poorly constructed and presented. Neighbour-joining trees provide a very crude inference of evolutionary relationships; maximum-likelihood trees are easily constructed with modern software and are preferred for the type of analysis conducted here. Furthermore, the sequence alignment algorithm and lengths of aligned sequences are not disclosed, no bootstrap values are shown, the branch length scales are not defined, and branch lengths are not proportional to evolutionary distance.

Response: Thank you for your kind comment. We have reconstructed the phylogenetic trees with the necessary annotation information.

  1. For the rhabdovirus tree, a larger set of viruses should be used to place the isolate and the relationships to other viruses in a suitable context. This is particularly important as the authors refer to the rhabdovirus as a “novel” virus. However, all of the sequences presented in the tree are from isolates assigned to the same rhabdovirus species (Siniperhavirus chuatsi). Indeed, the authors report the G protein sequence of the rhabdovirus isolate to share 98.16% amino acid sequence identity with the G protein of HSHRV strain C1207 which is well within the range of identify for isolates of the same virus. Furthermore, the authors do not include Chinese rice-field eel rhabdovirus (MH319839) in their analyses and this is also assigned to the same rhabdovirus species. As a minimum effort, the authors should include all of these isolates in their tree and use, as an outgroup, a virus assigned to another closely related rhabdovirus species. Eelpout rhabdovirus (KR612230; species Siniperhavirus zoarces) would be a suitable outgroup.  A larger tree placing the new isolate in the context of other alpharhaboviruses infecting fish and marine mammals would be preferred.

Response: Thank you for your kind suggestion. We have reconstructed the phylogenetic  tree according to your opinion, which contains more sequences of rhabdovirus isolates.

  1. Reference to rhabdovirus MS2021 as a “novel virus” should be removed throughout. It is simply another isolate of a previously known and described virus. 

Response: Thank you for your kind suggestion. We have removed the ‘novel virus’ in the whole MS.

  1. The use of virus nomenclature and taxonomic classification throughout does not conform to conventions established by the International Committee on Taxonomy of Viruses (ICTV).  Virus names should not be capitalised unless they include proper nouns. This applies both in the text and in the figure legends. Virus species names should be italicised and should comply with the most recent release of the ICTV master species list (MSL); see https://talk.ictvonline.org/taxonomy.  The authors will also find the following reference to the revised taxonomy of rhabdoviruses infecting fish and marine mammals useful: https://doi.org/10.3390/ani12111363.

Response: Thank you for your useful opinion. We have revised the virus name and taxonomic classification in the whole MS according to your suggestion.

Round 2

Reviewer 1 Report

No

Reviewer 4 Report

Thank you for your response. The manuscript is significantly improved and is now acceptance for publication.